# Synthesis of thia-Michael-Type Adducts between Naphthoquinones and *N*-Acetyl-*L*-Cysteine and Their Biological Activity

**DOI:** 10.3390/molecules27175645

**Published:** 2022-09-01

**Authors:** Gabriele Micheletti, Carla Boga, Chiara Zalambani, Giovanna Farruggia, Erika Esposito, Jessica Fiori, Nicola Rizzardi, Paola Taddei, Michele Di Foggia, Natalia Calonghi

**Affiliations:** 1Department of Industrial Chemistry ‘Toso Montanari’, Alma Mater Studiorum-Università di Bologna, Viale Del Risorgimento 4, 40136 Bologna, Italy; 2Department of Pharmacy and Biotechnology, University of Bologna, 40126 Bologna, Italy; 3Department of Chemistry ‘G. Ciamician’, Alma Mater Studiorum-Università di Bologna, Via Selmi 2, 40126 Bologna, Italy; 4Department of Biomedical and Neuromotor Sciences, Alma Mater Studiorum-Università di Bologna, Via Irnerio 48, 40126 Bologna, Italy

**Keywords:** *N*-acetyl-*L*-cysteine, cancer, naphthoquinone, juglone, thia-Michael, menadione, naphthazarin, plumbagin

## Abstract

A series of naphthoquinones, namely, 1,4-naphthoquinone, menadione, plumbagin, juglone, naphthazarin, and lawsone, were reacted with *N*-acetyl-*L*-cysteine, and except for lawsone, which did not react, the related adducts were obtained. After the tuning of the solvent and reaction conditions, the reaction products were isolated as almost pure from the complex reaction mixture via simple filtration and were fully characterized. Therefore, the aim of this work was to evaluate whether the antitumor activity of new compounds of 1,4-naphthoquinone derivatives leads to an increase in ROS in tumor cell lines of cervical carcinoma (HeLa), neuroblastoma (SH-SY5Y), and osteosarcoma (SaOS2, U2OS) and in normal dermal fibroblast (HDFa). The MTT assay was used to assay cell viability, the DCF-DA fluorescent probe to evaluate ROS induction, and cell-cycle analysis to measure the antiproliferative effect. Compounds **8**, **9**, and **12** showed a certain degree of cytotoxicity towards all the malignant cell lines tested, while compound **11** showed biological activity at higher IC_50_ values. Compounds **8** and **11** induced increases in ROS generation after 1 h of exposure, while after 48 h of treatment, only **8** induced an increase in ROS formation in HeLa cells. Cell-cycle analysis showed that compound **8** caused an increase in the number of G0/G1-phase cells in the HeLa experiment, while for the U2OS and SH-SY5Y cell lines, it led to an accumulation of S-phase cells. Therefore, these novel 1,4-naphthoquinone derivatives may be useful as antitumoral agents in the treatment of different cancers.

## 1. Introduction

Quinones are a class of organic compounds widely distributed in nature in animals, plants and microorganisms, where they are biosynthesized as secondary metabolites [1]. They can be chemically sub-divided into benzoquinones, naphthoquinones, and anthraquinones based on the carbocyclic aromatic ring of their corresponding reduced hydroquinonic form. The first two classes admit two isomeric forms, due to the relative position of the carbonyls within the ring system: 1,2-benzo- (or naphtho-) quinones and 1,4- benzo- (or naphtho-) quinones (Figure 1).

Quinones are highly reactive compounds of interest both in the mechanistic and applied fields [2,3]. For example, they can find application in natural and synthetic dyes [4,5], and biopesticides and bioherbicides [6,7] and play important roles in many biochemical processes, being involved in cellular respiration [8,9], photosynthesis [10], and cellular defense [11,12]. In the pharmaceutical field, they are used as trypanocidal, anti-inflammatory [13], antithrombotic [14,15], antiviral [16], antifungal [17], and antitumor agents [18], but their clinical use as antineoplastic drugs [19] has been limited for a long time by their toxicity.

Their biological actions are still difficult to be understood, although at least two competing mechanisms have been identified: (i) the ability to undergo redox cycling, generating reactive oxygen species (ROS), and (ii) their ability to act as electrophiles via Michael-type addition, producing covalent bonds via reactions with thiols or other nucleophiles present in proteins, DNA, and RNA [20].

Even if quinones and amino acids are usually compartmentally separated in living systems [21], there are several cases in which they meet and react. It occurs, for example, in wounded, cut, or crushed plant material during harvest, and in ensiling or disintegrating cells [21]. The reaction between quinonic compounds and amino acids takes place via 1,4-Michael-like addition with free nucleophilic functional groups such as sulfhydryl, amine, indole, and imidazole substituents. The formation of related conjugates (usually brown or black colored) influences the color, taste, and aroma of foods and is the reason for the pigmentation of various tissues.

Focusing attention on 1,4-naphthoquinone and some of its derivatives (Figure 2), mainly of natural occurrence, a few studies are reported about their reaction with amino acids, despite the presence in the literature of reports on their biological activities [20,21].

For example, recent studies have revealed that juglone (5-hydroxynaphthalene-1,4-dione, often also indicated as 5-hydroxy-1,4-naphthoquinone (**4**); Figure 2), a brown pigment deriving from black walnut (*Juglans regia* L.), influences cell signaling and is an inhibitor of peptidyl-prolyl cis/trans isomerase (Pin1) that can regulate the phosphorylation of Tau and activates mitogen-activated protein kinases that could promote cell survival, thus protecting against conditions such as cardiac injury [22]. Plumbagin (5-hydroxy-2-methylnaphthalene-1,4-dione, often also indicated as 5-hydroxy-3-methyl-1,4-naphthoquinone), mainly found in plants of the Plumbago family, as well as in some species of the *Ancestrocladaceae*, *Dioncophyllaceae*, *Droseraceae*, and *Ebenceae* families, induces cytotoxicity by modulating genes involved in angiogenesis and is used [23] together with other drugs in the treatment of chemo- and radio-resistant cancers. Both juglone and plumbagin have shown cytotoxicity to HaCaT keratinocytes, attributed to two different mechanisms, namely, redox cycling and reaction with glutathione (GSH) [24]. Naphthazarin (5,8-dihydroxynaphthalene-1,4-dione, often also indicated as 5,8-dihydroxy-1,4-naphthoquinone (**5**); Figure 2), a compound present in the *Boraginaceae*, *Droseraceae*, and *Nepenthaceae* families [25,26], has shown effects on oral squamous cell carcinoma [27], human breast cancer [28], human gastric cancer [29], and human colorectal cancer [30] cells. Menadione (2-methyl-1,4-naphthoquinone), also known as vitamin K_3_, is a synthetic compound used as a precursor in the synthesis of K_1_ and K_2_ vitamins and has showed to possess a wide range of biological activities, such as anticancer, antibacterial, antifungal, antimalarial, and anthelmintic activities [1].

Naphthoquinones are often characterized by poor bioavailability [23], and *N*-acetyl-*L*-cysteine, beside introducing amino acid cysteine in the new adducts, could improve their cellular uptake. Moreover, *N*-acetyl-*L*-cysteine, a derivative of cysteine, is an intermediary in the conversion of cysteine to GSH, and its sulfhydryl group can scavenge free radicals. It is readily hydrolyzed to cysteine and is able to expand natural antioxidant defenses by increasing reduced intracellular GSH concentration.

In the current study, we report the synthesis under mild conditions of adducts between some 1,4-naphtoquinones and *N*-acetyl-*L*-cysteine via thia-Michael-like addition, with some of them never having been reported so far.

Preliminary studies on their biological activity against a panel of cancer cell lines are also reported.

## 2. Results and Discussion

### 2.1. Reactions between 1,4-Naphthoquinones and N-Acetyl-L-Cysteine

The reaction between naphthoquinones **1**–**6** and *N*-acetyl-*L*-cysteine (**7**) was first carried out without catalysts and under mild conditions to mimic the situation occurring during the interaction of quinones with thiol residues of proteins (enzymes, keratin, or animal fibers previously treated with reducing agents). Moreover, cysteine protected at the amino group was used to avoid cyclization through the formation of a Schiff base with a carbonyl group of the quinone [31].

The reaction course can occur as depicted in Figure 1; the first step is thia-Michael 1,4-addition to the β−position of the α,β−unsaturated quinonic conjugated system giving intermediate **A**, which quickly evolves to neutral species **B**. The latter undergoes tautomerism to produce hydroquinonic form **C**, owing to the energy gained due to the aromaticity.

This is in agreement with the literature reports on the reaction between mercaptans and 1,4-benzoquinone where the isolated products were in hydroquinonic form [32].

However, in general, hydroquinonic forms can easily be oxidized to quinonic ones. This can occur via air oxidation or during the reaction where the presence of a not-yet-reacted quinone can cause the oxidation of the product with the contemporary reduction of the starting quinone, with consequent lack of its reactivity as Michael acceptor and lowering of the yields (see Figure 2, Path. A, applied to the 1,4-benzoquinone case) [2,3,32].

Moreover, *N*-acetyl-*L*-cysteine can also be oxidized to a cystinic derivative by the addition product in quinonic form (Path. B) or by the starting quinone (Path. C); this can occur especially when the reaction is particularly slow, as in the case of not-catalyzed reactions.

As it can be evinced, Michael-like 1,4-addition to quinonic acceptors can be quite complicated. The reactions herein reported are shown in Figure 3.

They were carried out under different experimental conditions.

At first, the reaction of **7** with **2**, **3**, **4**, or **5** was carried out in methanol at room temperature; after about 5 h, the ^1^H NMR spectrum of the concentrated crude reaction mixture revealed the presence of starting materials together with numerous signals, some of them ascribable to reaction products. We tried to isolate the products via column chromatography on silica gel, but likely due to decomposition phenomena, we obtained yields lower than those expected on the basis of the ^1^H NMR of the crude, and in the case of **12**, no products were isolated.

As for juglone (**4**), after chromatographic purification, a mixture of the two regioisomers, **11** and **11a**, in a 7:3 relative ratio was isolated (Figure 4).

Compounds **11** and **11a** have not been reported so far and were characterized using ^1^H- and ^13^C-NMR, and mass spectrometry. In particular, the signals of the two regioisomers were attributed on the basis of *g*-HMBC and *g*-HSQC experiments.

The reaction with lawsone (**6**) did not occur, possibly because of the presence, not in a negligible amount, of the tautomeric keto form [33] (Figure 3), which deactivates the nucleophilic attack on **6**.

To overcome the difficulties to isolate the products from the complicated reaction mixture, we tried to repeat the reactions with a different solvent. When the reactions were carried out in ethanol at 25 °C, surprisingly, in the cases of the reaction with **1**–**3,** we noted the precipitation of a solid in the reaction mixture whose ^1^H NMR spectrum corresponded to that of products **8**–**10** and **12**, respectively. This represented a very important finding that permitted to isolate the product via simple filtration, avoiding the purification step through column chromatography. From the reaction with quinone **4,** no precipitation was observed in ethanol, and a mixture of **11** and **11a** was isolated using column chromatography on silica gel. The reactions in ethanol were also carried out at 50 °C. As shown in Table 1, an increase in the reaction temperature gave a noticeably increase in the yield of precipitated product.

The analyses of compounds **8**–**12** isolated via filtration revealed that they were of purity suitable for biological studies; obviously, the yields might increase after an optimization of the reaction conditions, but at this time, this was not our objective.

Further, some reactions were also carried out in acetonitrile; from the reaction between **1** and **7** at 50 °C, after 3h, compound **8** was separated via precipitation with a 33% yield.

From the reaction between menadione and **7**, kept in acetonitrile at 50 °C and then under reflux, no products precipitated, and the ^1^H NMR spectrum showed negligible conversion to **9,** whereas lawsone did not react under the above conditions.

Finally, from the reaction between juglone **4** and **7** in acetonitrile at room temperature, a precipitate was separated, and it resulted to be compound **11** (only traces of **11a** were detected via ^1^H NMR). This finding is of particular interest, since this method permits one to isolate only one, almost pure regioisomer. The reaction was reproducible, and the structure of **11** was ascertained using 2D NMR experiments. This finding is of particular interest in light of contrasting results previously reported on the regioselectivity of the reaction between thiophenol and juglone [34].

### 2.2. Biological Activity

Molecules **8**–**12** were tested *in vitro* using the MTT assay against four cancer cell lines: HeLa, SH-SY5Y, SaOS2, and U2OS. The substances that displayed significant results against the cancer cell lines were also investigated against a normal cell line, HDFa, to evaluate their selective activity. Cytotoxic activities were expressed as IC_50_ values, i.e., the concentration of the substance that reduced cell viability by 50% (Table 2).

The results presented in Table 2 and Figure 4 reveal that **8**, **9**, and **12** exhibited some degree of cytotoxicity against all the malignant cell lines tested, with IC_50_ values in the range of 0.50–1.81 μM. On the contrary, **11** showed biological activity with much higher IC_50_ values. In HDAFa, **12** exhibited biological activity with an IC_50_ very similar to that found in all tumor cell lines, while **11** had a lower IC_50_ than that calculated in tumor cell lines. Among the substances of the series, **10** was inactive against all the cancer cell lines examined and was not tested against HDFa.

SaOS2 and U2OS are human osteosarcoma cell lines, but while SaOS2 cells represent a mature phenotype, U2OS cells are negative for almost all osteoblastic markers but positive for cartilage markers such as collagen II, IX, and X. Additionally, U2OS cells show positive results for type IV collagen, which is expressed only in very early differentiation stages but not by mature osteoblasts [35,36]. For this reason, we decided to continue the studies on the biological effects of our molecules using HeLa, U2OS, and SH-SY5Y as models and **8** (which had a lower IC_50_) and **11** (which had a higher IC_50_) as molecules to be administered.

#### 2.2.1. Redox Imbalance

It is known that non-vitamin K naphthoquinones of natural origin, such as plumbagin or juglone, are excellent redox cyclers, causing the generation of ROS in cells exposed to these quinones [24,37]. In the case of naphthoquinones, redox cycling represents a cyclic process of the reduction of a compound, followed by the (auto-) oxidation of the reaction product under the concomitant generation of ROS. In the case of 1,4-naphthoquinones and mammalian cells, the reduction of these quinones may occur at the expense of NADH or NADPH, as catalyzed by several alternate enzymes. For example, cytochrome P450 reductase can catalyze the simple reduction of naphthoquinone to the corresponding semiquinone [38] or, alternatively, may undergo reduction to the corresponding hydroquinone, catalyzed by NAD(P)H:quinone oxidoreductase-1 (NQO-1, DT-diaphorase) [39].

To further explore and establish a direct relationship between **8** or **11** and ROS, DCFH-DA staining was applied to HeLa, U2OS, and SH-SY5Y cells treated for 1 h and 48 h. The results indicated that **8** and **11** induced increases in ROS generation after 1 h of exposure, but the biological response was different in the three cell lines, as shown in Figure 5A. Indeed, in HeLa cells, **8** induced an increase in ROS formation of 1.14 ± 0.042, while in U2OS and SH-SY5Y cells, the increases were 1.30 ± 0.13-fold and 1.51 ± 0.16-fold with respect to control, with *p* < 0.0001, *p* < 0.0005, and *p* < 0.0024, respectively. In the same way, **11** induced increases in ROS formation in HeLa cells of 1.27 ± 0.16, while in U2OS and SH-SY5Y cells, the increases were 1.62 ± 0.11-fold and 1.61 ± 0.14-fold with respect to control, with *p* < 0.0049, *p* < 0.0001, and *p* < 0.0056, respectively. Moreover, it is important to note that both compounds **8** and **11** induced a greater formation of ROS in U2OS and SH-SY5Y cells than in HeLa cells, as shown in Figure 5A.

After 48 h of treatment with **8** or **11,** there was no ROS production in U2OS nor SH-SY5Y cells. Interestingly, only **8** induced an increase in ROS formation in HeLa cells of 2.28 ± 0.47-fold with respect to control (*p* < 0.012) (Figure 5B).

#### 2.2.2. Effect on Cell Proliferation

The data reported in Figure 5 show how both **8** and **11** induced ROS formation in a similar way, but **8** showed a much lower IC_50_. For this reason, we decided to analyze the effects on proliferation by treating the cells only with **8**. Next, we explored the effect of 5 μM **8** on cell-cycle distribution in HeLa, U2OS, and SH-SY5Y cancer cells using PI stain followed by flow cytometry. Cells were exposed to the compound for 24 or 48 h prior to processing and analysis.

As shown in Figure 6A,B(above), in the HeLa experiment, treatment with **8** resulted in an increase in the number of G0/G1-phase cells of 1.3% with respect to control, while for the U2OS and SH-SY5Y cell lines, it resulted in increases in S-phase cells of 1.3% and 1.1%, respectively.

After 48 h of treatment with **8** in HeLa, there was an increase in the number of cells in the S phase of 1.2% with respect to control, while in U2OS and SH-SY5Y cells, there were still accumulations of cells in the S phase of 1.1% and 1.3% with respect to control, respectively, as shown in Figure 6A,B(below).

In conclusion, **8** induced an ROS-mediated antiproliferative effect in HeLa, U2OS, and SH-SY5Y cells, which modulated the cell-cycle distribution. Interestingly, the *in vitro* potency to increase radical formation and induce biological effects strongly depended on the molecular biological characteristics of the individual cell lines.

#### 2.2.3. Product Reactivity with Glutathione

Naphthoquinones may react with nucleophiles, such as thiols or amines, and form adducts via the so-called Michael addition reaction [37,40]. Furthermore, naphthoquinones interact with nucleophiles, such as glutathione, causing significant GSH modification and thus depletion in cells exposed to these compounds [37,41,42].

To understand if both the different formation of radicals and the biological response induced by **8** observed in the three cancer models can actually be traced back to a different molecular structure, we decided to conclude this study by verifying whether these molecules can form adducts with GSH.

To this end, **8**, **9**, and **11** were incubated with GSH in 0.1 mL phosphate-buffered saline (PBS) overnight; then, these solutions underwent direct-infusion mass spectrometric analyses (DI-MS). The mass spectra (Appendix A) showed the formation of two main adducts of compounds **8** and **11** with glutathione and with *N*-acetyl-*L*-cysteine, and other reaction products, such as the dimers of *N*-acetyl-*L*-cysteine and glutathione (GSSG) and the heterodimer between *N*-acetyl-*L*-cysteine and GSH. Furthermore, the reaction solution containing compound **8** showed the adduct formation between naphthoquinone (**1**) and GSH. Otherwise, **9** (Appendix A) did not seem to react with GSH; in fact, in the spectrum of the reaction mixture, we only found adducts *N*-acetyl-*L*-cysteine—GSH, menadione (**2**)—GSH, and GSSG. The DI-MS analyses of the reaction mixtures showed that **8** and **11** reacted with GSH and *N*-acetyl-*L*-cysteine, leading to a decrease in the amount of the starting compounds; on the other hand, **9** remained almost unaffected.

## 3. Materials and Methods

### 3.1. Chemical Synthesis

#### 3.1.1. General

Nuclear magnetic resonance spectra (^1^H NMR and ^13^C NMR,) were recorded at 25 °C on Varian INOVA 600 spectrometers (Varian, Palo Alto, CA, USA) operating at 600 MHz (for ^1^H NMR) and 150.80 MHz (for ^13^C NMR). Chemical shifts were referenced to the solvent (^1^H NMR, 3.31 ppm for CD_3_OD and 2.50 ppm for DMSO-d_6_; ^13^C NMR, 49.00 ppm for CD_3_OD and 39.50 ppm for DMSO-d_6_). J values were given in Hz. Chromatographic purifications (FCs) were carried out on glass columns packed with silica gel Geduran Si 60, 0.063–0.200 mm (Sigma-Aldrich, Milan, Italy) at medium pressure. Thin-layer chromatography (TLC) was performed on aluminum foils coated in silica gel 60 F254 (Fluka, Buchs, Switzerland). Melting points were recorded on a Büchi apparatus (Stone, Staffs, UK). ESI-MS spectra were obtained with a WATERS 2Q 4000 instrument (Waters Corporation, Milford, MA, USA).

#### 3.1.2. General Procedure

In a three-necked flask equipped with a circular condenser, we introduced quinone (1 mmol) and *N*-acetyl-*L*-cysteine (1 mmol) in 10 mL of ethanol (or acetonitrile). The mixture was kept at room temperature or heated to 50 °C under magnetic stirring. The course of the reaction was monitored using TLC analyses or ^1^H-NMR spectroscopy. During the reaction’s course, a solid was formed. At the end of the reaction, the solid was separated via filtration and washed with cold ethanol (or acetonitrile). The obtained solid was led to dryness *in vacuum*. If not otherwise specified, the yields given below were obtained from the reaction in ethanol at 50 °C.

*N-acetyl-S-(1,4-dioxo-1,4-dihydronaphthalen-2-yl)-L-cysteine* (**8**)

Yellow solid; m.p., 220.3–221.7 °C; yield, 34%. ^1^H-NMR: (CD_3_OD, 600 MHz, 25 °C): δ, ppm: 8.08 (dd, ^1^*J* = 7.6 Hz, ^2^*J* = 1.5 Hz, 1H), 8.06 (dd, ^1^*J* = 7.6 Hz, ^2^*J* = 1.4 Hz, 1H) 7.82 (dd, ^1^*J* = 7.6 Hz, ^2^*J* = 1.5 Hz, 1H), 7.78 (dd, ^1^*J* = 7.6 Hz, ^2^*J* = 1.4 Hz, 1H) 6.84 (s, 1H), 4.78 (dd, ^1^*J* = 8.4 Hz, ^2^*J* = 4.6 Hz, 1H), 3.50 (dd, ^1^*J* = 13.6 Hz, ^2^*J* = 4.6 Hz, 1H), 3.26 (dd, ^1^*J* = 13.6 Hz, ^2^*J* = 8.4 Hz, 1H), 2.00 (s, 3H). ^13^C-NMR: (CD_3_OD, 150.0 MHz, 25 °C): δ, ppm: 183.1 (C), 182.7 (C), 173.5 (C), 172.8 (C), 155.3 (C), 135.6 (CH), 134.6 (CH), 133.5(C), 133.3 (C), 128.6 (CH), 127.6 (CH), 127.4 (CH), 52.1 (CH), 29.2 (CH_2_), 22.3 (CH_3_). C_15_H_13_NO_5_S, M.W.: 319.33. ESI-MS (ESI^–^): *m*/*z*: 318 [M-H]^–^.

*N-acetyl-S-(3-methyl-1,4-dioxo-1,4-dihydronaphthalen-2-yl)-L-cysteine* (**9**)

Dark yellow solid; m.p., 206.9–208.3 °C; yield, 25%. ^1^H-NMR: (CD_3_OD, 600 MHz, 25°C): δ, ppm: 8.10–8.05 (m, 2H), 7.79–7.76 (m, 2H), 4.62 (dd, ^1^*J* = 7.74 Hz, ^2^*J* = 4.57 Hz, 1H), 3.76 (dd, ^1^*J* = 14.21 Hz, ^2^*J* = 4.60 Hz, 1H), 3.45 (dd, ^1^*J* = 14.21 Hz, ^2^*J* = 7.60 Hz, 1H), 2.34 (s, 3H), 1.84 (s, 3H). ^13^C-NMR: (CD_3_OD, 150.0 MHz, 25 °C): δ, ppm: 183.4 (C), 182.2 (C), 173.1 (C), 173.0 (C), 149.5 (C), 146.5 (C), 134.9 (CH), 134.7 (CH), 134.3 (C), 133.5 (C), 127.8 (CH), 127.4 (CH), 54.6 (CH), 36.0 (CH_2_), 22.3 (CH_3_), 15.5 (CH_3_).C_16_H_15_NO_5_S, M.W.: 333.36. ESI-MS (ESI–): *m*/*z*: 332 [M-H]^–^.

*N-acetyl-S-(8-hydroxy-3-methyl-1,4-dioxo-1,4-dihydronaphthalen-2-yl)-L-cysteine* (**10**).

Dark purple solid; m.p. >193 °C dec; yield, 40%. ^1^H-NMR: (CD_3_OD, 600 MHz, 25 °C): δ, ppm: 7.62 (t, ^1^*J* = 8.97 Hz, 1H), 7.56 (d, *J* = 7.69 Hz, 1H), 7.22 (d, *J* = 8.33 Hz, 1H), 4.47–4.43 (m, 1H), 3.84 (dd, ^1^*J* = 13.64 Hz, ^2^*J* = 3.41,1H), 3.43 (dd, ^1^*J* = 13.64 Hz, ^2^*J* = 7.67 Hz, 1H), 2.33 (s, 3H), 1.82 (s, 3H). ^13^C-NMR: (CD_3_OD, 150.0 MHz, 25 °C): δ, ppm: 187.5 (C), 182.7 (C), 176.4 (C), 172.8 (C), 162.7 (C), 150.7 (C), 116.8 (C), 137.3 (CH), 133.7 (C), 124.7 (CH), 119.9 (CH), 116.8 (C), 56.8 (CH), 37.6 (CH_2_), 22.6 (CH_3_), 15.8 (CH_3_). C_16_H_15_NO_6_S, M.W.: 349.36. ESI-MS (ESI^–^), *m*/*z*: 348 [M-H]^–^.

*N-acetyl-S-(5-hydroxy-1,4-dioxo-1,4-dihydronaphthalen-2-yl)-L-cysteine* (**11**).

Orange solid precipitated from acetonitrile; m.p., 209.8–211.7 °C; yield, 38% from acetonitrile at 25 °C. ^1^H-NMR: (DMSO-d_6_, 600 MHz, 25 °C): δ, ppm: 13.12 (br.s, 1H), 12.06 (s, 1H), 8.49 (d, *J* =7.9 Hz, 1H), 7.71 (t, *J* = 8.2Hz, 1H), 7.55 (d, *J* = 7.6 Hz, 1H), 7.36 (d, *J* = 8.7 Hz, 1H), 6.84 (s, 1H), 4.59-4.53 (m, 1H), 3.38 (dd, ^1^*J* = 13.5 Hz, ^2^*J* = 5.0 Hz, 1H), 3.22 (dd, ^1^*J* = 13.5 Hz, ^2^*J* = 8.1 Hz, 1H) 1.86 (s, 3H). ^13^C-NMR: (DMSO-d_6_, 150.0 MHz, 25 °C): δ, ppm: 186.5 (C), 181.1 (C), 171.3 (C), 169.6 (C), 160.4 (C), 154.7 (C), 136.1 (CH), 131.5 (C), 127.1 (CH), 124.8 (CH), 119.2 (CH), 114.3 (C), 50.3 (CH), 31.4 (CH_2_), 22.3 (CH_3_). C_15_H_13_NO_6_S, M.W.: 335.33. ESI-MS (ESI–): m/z: 334 [M-H]^–^.

*N-acetyl-S-(5-hydroxy-1,4-dioxo-1,4-dihydronaphthalen-2-yl)-L-cysteine* (**11**) *and N-acetyl-S-(8-hydroxy-1,4-dioxo-1,4-dihydronaphthalen-2-yl)-L-cysteine* (**11a**).

Yellow solid containing **11** + **11a** isolated using column chromatography of the crude from reaction in methanol or ethanol; ^1^H-NMR: (CD_3_OD, 600 MHz, 25 °C): mixture of **11**:**11a** in 7:3 relative molar ratio, δ, ppm: 7.67 (t, *J* = 8.4 Hz, 1H, **11a**), 7.61 (t, *J* = 7.75 Hz, 1H, **11**), 7.58 (t, *J* = 7.52 Hz, 1H, **11**), 7.54 (dd, *J* = 7.6 Hz, *J* = 1.2 Hz, 1H, **11a**), 7.25 (dd, *J* = 8.0 Hz, *J* = 1.4 Hz, 1H, **11**), 7.22 (dd, *J* = 8.4 Hz, *J* = 1.2 Hz, 1H, **11a**), 6.85 (s, 1H, **11** + **11a**), 4.66-4.59 (m, 1H, CHN, **11** + **11a**), 3.49 (dd, *J* = 13.2 Hz, *J* = 4.5 Hz, 1H, **11** + **11a**), 3.24 (d, *J* = 12.9 Hz, *J* = 7.4 Hz, 1H, **11**), 3.23 (d, *J* = 13.1 Hz, *J* = 7.6 Hz, 1H, **11a**), 2.01 (s, 3H, **11**), 1.98 (s, 3H, **11a**). ^13^C-NMR: (CD_3_OD, 150.0 MHz, 25 °C): δ, ppm: 188.5 (C), 182.8 (C, **11**), 182.5 (C, **11a**), 176.1 (C), 173.4 (C), 162.7 (C), 157.8 (C, **11**), 156.3 (C, **11a**), 139.9 (CH, **11**), 138.3 (CH, **11a**), 133.8 (C, **11a**), 133.5 (C, **11**), 129.2 (CH, **11a**), 128.2 (CH, **11**), 125.8 (CH, **11**), 124.7 (CH, **11a**), 120.5 (CH, **11**), 120.1 (CH, **11a**), 116.3 (C, **11a**), 116.0 (C, **11**), 56.3 (CH, **11** + **11a**), 34.78 (CH_2_, **11**), 34.72 (CH_2_, **11a**), 22.9 (CH, **11** + **11a**).

*N-acetyl-S-(5,8-dihydroxy-1,4-dioxo-1,4-dihydronaphtalen-2-yl)-L-cysteine* (**12**).

Dark purple solid; m.p., 238.2-239.9 °C; yield, 13%. ^1^H-NMR: (DMSO-d_6_, 600 MHz, 25 °C): δ, ppm: 13.16 (br.s, 1H), 12.55 (s, 1H), 11.82 (s, 1H), 8.49 (d, *J* = 8.2 Hz, 1H) 7.41 (d, *J* = 9.5 Hz, 1H), 7.35 (d, *J* = 9.4 Hz, 1H), 6.91 (s, 1H), 4.60-4.50 (m, 1H), 3.41 (dd, ^1^*J* = 13.6 Hz, ^2^*J* = 4.3 Hz, 1H); 3.24 (dd, ^1^*J* = 13.6 Hz, ^2^*J* = 8.0 Hz, 1H), 3.55-3.40 (m, 1H), 1.86 (s, 3H). ^13^C-NMR: (DMSO-d_6_, 150.0 MHz, 25 °C): δ, ppm: 184.1 (C), 183.5 (C), 171.3 (C), 169.6 (C), 157.5 (C), 157.0 (C), 154.3 (C), 130.5 (CH), 128.7 (CH), 127.5 (CH), 111.6 (C), 111.1 (C), 50.3 (CH), 31.3 (CH_2_), 22.3 (CH_3_). C_15_H_13_NO_7_S, M.W.: 351.33. ESI-MS (ESI^–^): *m*/*z*: 350 [M-H]^–^.

### 3.2. Biology

#### 3.2.1. Cell Culture and Treatments

HeLa, SH-SY5Y, SaOS2, U2OS, and HDFa cell lines were purchased from American Type Culture Collection (ATCC; Manassas, VA, USA). HeLa, SaOS2, and U2OS cells were cultured in RPMI 1640 medium (Labtek Eurobio, Milan, Italy), while SH-SY5Y and HDFa cells were cultured in high-glucose DMEM (Labtek Eurobio, Milan, Italy), and both were supplemented with 10% FCS (Euroclone, Milan, Italy) and 2mM L-glutamine (Sigma-Aldrich, Milan, Italy), at 37 °C in a 5% CO_2_ atmosphere. The compounds were dissolved in DMSO in a 30–50 mM stock solution. In cell treatments, the final DMSO concentration never exceeded 0.1%.

#### 3.2.2. MTT Assay

Cells were seeded at 1.5 × 10^4^ cells/well in a 96-well culture plastic plate (Orange Scientific, Braine-l’Alleud, Belgium), and after 24 h of growth, they were exposed to increasing concentrations of compounds (from 0.01 to 200 μM) solubilized in RPMI 1640 medium. After 48 h of treatment, the culture medium was replaced with 0.1 mL of 3-(4,5-dimethylthiazolyl-2)-2,5-diphenyltetrazolium bromide (MTT; Sigma-Aldrich, Milan, Italy) dissolved in PBS at the concentration of 0.2 mg/mL, and samples were incubated for 2 h at 37 °C. The formazan salt crystals formed were dissolved with 0.2 mL of isopropyl alcohol for 20 min. The absorbance at 570  nm was measured using a multi-well plate reader (Tecan, Männedorf, Switzerland), and data were analyzed using Prism GraphPad software and expressed as IC_50_.

#### 3.2.3. Measurement of Intracellular ROS Level

Oxidative stress was measured in intact cells using an ROS indicator, 2′,7′-dichlorodihydrofluorescein diacetate (DCFDA; Thermo Fisher Scientific, Waltham, MA, USA). HeLa, U2OS, and SH-SY5Y cells were seeded in 96-well plates at 1.5 × 10^4^ cells/well (Optiplate, Perkin Elmer, Milan, Italy), and after 24 h to allow adhesion, HeLa and SH-SY5Y cells were incubated with 5 μM GA 462 or 20 μM SM71, while U2OS were treated with 5 μM **8** or 30 μM **11**, dissolved in complete medium for 1 or 48 h at 37 °C in 5% CO_2_ atmosphere. After this time, cells were washed with PBS and treated with 10 μM DCFDA in culture medium for 30 min. Finally, cells were washed again with PBS and the fluorescence value in each well was measured (λexc = 488 nm; λem = 530 nm) with a plate reader (Tecan, Männedorf, Switzerland). Fluorescence emission was normalized to protein content measured using the Bradford method.

#### 3.2.4. Cell-Cycle Analysis via Flow Cytometry

The effects on the HeLa and U2OS cell cycles were studied after 24 or 48 h of treatment with 5 μM **8**. Cells were detached with 0.11% trypsin, washed in PBS and centrifuged. The pellet was resuspended in 0.01% Nonidet P-40, 10 μg/mL RNase, 0.1% sodium citrate, and 50 μg/mL PI, for 30 min at room temperature in the dark. Flow cytometric analyses were performed as previously described [43].

#### 3.2.5. Reactions with GSH and DI-MS Analysis

The standard reaction mixture consisted of 3 mM **8**, **9**, and **11** in 0.1 mL PBS, pH 7.4 or 4.0; the reaction was started with the addition of 3 mM GSH, overnight at 37 ° C. Reaction mixtures, diluted at 1:100 in methanol, were analyzed in direct-infusion mode and negative polarity with an AB Sciex QTrap 4500 (Concord, ON, Canada) mass spectrometer. The experimental conditions were as follows: ion spray voltage—4500 kV; temperature—100 °C; curtain gas (nitrogen) pressure—30 PSI; declustering potential—60 V; entrance potential—10 V. Mass spectra were acquired with Analyst 1.6.3 software (AB Sciex, Concord, ON, Canada).

#### 3.2.6. Statistical Analysis

Statistical analyses were performed with GraphPad Prism software.

All data were expressed as means ± SDs. Student’s test was performed for the biological experiments; statistical differences were considered significant at a value of *p* < 0.05 and were reported as *p* < 0.01 (**) and *p* < 0.001 (***).

## 4. Conclusions

The thia-Michael-like addition of *N*-acetyl-*L*-cysteine to 1,4-naphthoquinone (**1**), menadione (**2**), plumbagin (**3**), juglone (**4**), or naphthazarin (**5**) followed by in situ oxidation of the hydroquinonic form to naphthoquinone derivatives **8**–**12** was carried out under different experimental conditions.

By changing the solvent, the conditions to obtain the products as precipitates from the complex reaction mixture were found. This method, despite lowering the yield of the recovered product, permitted us to separate it via simple filtration, avoiding the chromatographic purification step often involving decomposition phenomena. The reaction with juglone in methanol or ethanol gave an inseparable mixture of regioisomeric compounds **11** and **11a** in a 7:3 relative molar ratio, whereas in acetonitrile at room temperature, compound **11** precipitated and was isolated via filtration; the products obtained (lawsone did not react under our experimental conditions) were characterized using ^1^H-NMR, ^13^C-NMR, and mass spectrometry.

1,4-naphthoquinone derivatives had potent cytostatic effects particularly against HeLa, SH-SY5Y, SaOS2, and U2OS cancer cell lines based on IC_50_ values. In particular, compounds **8**, **9**, and **11** were also more selective for these tumors than HDFa. Compounds **8** and **11** induced increases in ROS generation after 1 h of exposure, while only **8** induced an increase in ROS formation in HeLa cells after 48 h of treatment. Furthermore, cell-cycle analyses showed that compound **8** caused an increase in the number of G0/G1-phase cells in the HeLa experiment, while for the U2OS and SH-SY5Y cell lines, it led to an accumulation of S-phase cells.

Interestingly, the *in vitro* potency to increase radical formation and induce biological effects could depend on the molecular biological characteristics of individual cell lines, as, for example, in glutathione disulfide reductase (GSR) and NQO1 content. In fact, by consulting the atlas of human proteins (https://www.proteinatlas.org/ (accessed on 20 August 2022)), it is possible to see how the gene expression of GSR and NQO1 is very different between these cell lines: in the HeLa cell line, GSR corresponds to 67.0 normalized transcript expression values (nTPM), while in the U2OS and SH-SY5Y cell lines, it corresponds to 34.7 and 31.7 nTPM, respectively; in the HeLa cell line, NQO1 corresponds to 564.4 nTPM, while in the U2OS and SH-SY5Y cell lines, it corresponds to 155.8 and 17.2 nTPM, respectively.

The important information obtained in this work was given by the mass analyses of the reaction mixture containing the derivatives of naphthoquinone and GSH. In fact, the mass spectra of the reaction solution containing compound **8** showed adduct formation between naphthoquinone and GSH, while **9** did not seem to react with GSH. In fact, in the spectrum of the reaction mixture, we only found adducts *N*-acetyl-*L*-cysteine—GSH, menadione—GSH, and GSSG. The DI-MS analyses of the reaction mixtures showed that **8** and **11** reacted with GSH and *N*-acetyl-*L*-cysteine, leading to a decrease in the amount of the starting compounds, while **9** remained almost unaffected.

The results suggested a role of GSH adducts in the observed biological effects on cancer cells.

## Data Availability

Not applicable.

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
