# Peer review of "Synthesis of thia-Michael-Type Adducts between Naphthoquinones and N-Acetyl-L-Cysteine and Their Biological Activity"

_molecules, 2022, doi:10.3390/molecules27175645_

Round 1

Author Response

Point to Point Response to Reviewers´ Comments on Manuscript Molecules - 1876580

A detailed response to the comments and suggestions of reviewer’s 1, that we sincerely thank, is given below. Therein, at first the original referee´s comment is shown in black colour followed by author’s response in blue.

All revisions made to the manuscript have been yellow highlighted.

REVIEWER 1 comments:

In this original research paper, the authors synthesized and characterized a couple of 1,4- naphthoquinione derivatives through Michael-like addition with N-acetyl-L-cysteine (NAC). In addition, they evaluated the biological activity of those derivatives through cytotoxicity assay, determination of intracellular ROS level, cell-cycle analysis and analysis of reactivity with glutathione. The reviewer considers that this manuscript provide a topic of interest to the audiences in this field with proven evidence on the production of 1,4-naphthoquinione derivatives and evaluation of their biological activity. However, major revision of the content is required before consideration of its publication at Molecules.

  1. In the abstract session (line 22), the reviewer suggests the addition of cervical carcinoma and neuroblastoma cell lines used in the paper to be consistent with osteosarcoma.

Answer: We thank the reviewer for the suggestion, and we change the manuscript as required.

  1. In the introduction session, 1) chemical names of naphthoquinones (line 72 ~ 88) are not consistent with the same compound in Figure 2;

Answer: Chemical names given in Figure 2 are IUPAC names, we added these in the main text specifying that also the other name written in the first version is often used.

2) in line 87, be specific with what cancers are affected by naphthazarin;

Answer: we specified what cancers are affected, changing the sentence as follows:

Naphthazarin ….showed effect on oral squamous cell carcinoma [27], human breast cancer [28], human gastric cancer [29], and human colorectal cancer [30] cells.

and 3) add more content to discuss why to investigate the effect of adducts between naphthoquinone and NAC.

Answer: we added the following sentences in the introduction:

“Naphtoquinones are often characterized by poor bioavailability [23], and N-acetyl-L-cysteine, beside to introduce the aminoacid cysteine in the new adducts, could improve their cellular uptake. Moreover, N-acetyl-L-cysteine a derivative of cysteine, is an intermediary in the conversion of cysteine to GSH, and its sulphydryl group can scavenge free radicals. It is readily hydrolyzed to cysteine and is able to expand natural antioxidant defenses by increasing intracellular reduced GSH concentration.”

  1. In the 2.1. session of results and discussion, 1) the reviewer suggests to add structure of NAC in Figure 2 so that audience knows what the structure of NAC in 7 refers to (line 99);

Answer: we thank the reviewer for the suggestion and added structure of NAC in Figure 2.

2) in scheme 2 and 3, NAC should be written as “N-acetyl-L-cysteine” NOT “N-acetyl-Lcystein”;

Answer: we thank the reviewer and corrected.

3) in line 155, it should be 25 or 50 degree?;

Answer: we thank the reviewer and corrected: the temperature is 25 °C and in cases 1-3 we noted precipitation.

and 4) table 1 is confusing especially where a stands. The reviewer highly suggests fix table 1 with better interpretation.

Answer: we thank the reviewer and tried to better explain Table 1 content.

  1. In the 2.2. session of results and discussion, 1) the reviewer suggests to present the cytotoxicity data of molecules 8 – 12 in bar graphs in addition to table 2; 2) in line 183, it should be molecule “8-12” NOT “1-4” tested;

Answer: done

and 3) in line 195, add “cell lines” after “in all tumor”; add “cell” after “tumor”.

Answer: done

  1. In the 2.3. session of results and discussion, 1) in line 211, “(aut)” should be “(auto)”;

Answer: done

 2) when check the relationship between 8/11 and ROS, why checking 48 hours? It’s usually better to observe the production of ROS at early time point like 3, 6, or 8 hours

Answer: We chose short and long times because the biological effects on cell cycle are still present also at 48 h.

and 3) in figure 4, name the graph on the left A and the one on the right B, and add a control group with no treatment in the bar graphs.

Answer: We changed

  1. IN the session of materials and methods, 1) under 3.2.4., which assay kit was used for the cell-cycle analysis by flow cytometry?

Answer: No kit was used but we stained the cells as reported in the text.

 2) under 3.2.6., which statistical software was used?

Answer: GraphPad Prism

  1. In the conclusion session, what does the results of different cell cycle in different cancer cells after the treatment of compound 8 tell us?

Answer: comments on this point are already included in the text (paragraph 2.2.2).

  1. Grammar and spelling errors: the reviewer strongly request a proofread of this paper because of many errors found in the paper, which are exemplified as follows. 1) line 66, “some its derivatives” should be “some of its derivatives”; 2) line 77, “manly” should be “mainly”; 3) line 79, “citotoxicity” should be “cytotoxicity”; 4) line 98, “cystein” should be “cysteine”; 5) line 155, “because permitted..” should be “because it permitted..”; 6) line 170, “..50 °C after 3h compound 8..” should be “..50 °C. After 3h compound 8..”; 7) line 183 and 250, “in vitro” should be “in vitro”; 8) line 186, “selectivity activity” should be 2 “selective activity”; 9) line 358, “(HDFa” should be “(HDFa)”; and 10) line 364, “0,1%” should be “0.1%”.

Answer: we thank the reviewer and corrected grammar and spelling errors.

  1. Check the abbreviation of PBS and ROS. Phosphate buffer saline (PBS) and reactive oxygen species (ROS) appear more than one time.

Answer: Done

Reviewer 2 Report

Micheletti, Calonghi et al. reported the naphthoquinones-cysteine conjugated compounds synthesis in different experimental conditions and tested their anticancer properties against various cancer cell lines. Compounds, particularly 8-12, have been characterized with the help of spectroscopic techniques such as NMR (1H and 13C) and ESI-MS. The correlation between the increase in reactive oxygen species (ROS) with the help of H2DCFDA and cell cycle analysis to measure the anti-proliferative effect has been evaluated. The author also assisted the toxic impact of respective naphthoquinones-cysteine 8-12 using a non-cancerous human dermal fibroblast (HDFa) cell line. Finally, the observed biological effects were investigated using glutathione, known for forming GSH adducts with RNA/DNA. The mass analysis of the reaction mixture of 8 showed adduct formation, whereas compound 9 remained almost unaffected.

The current manuscript version needs revision before acceptance for publication in the Molecule journal.

1. Many abbreviations are mentioned in the abstract, such as ROS, MTT, DFCDA, IC50, and SH-SY5Y, which need to be defined in the first place, including in the introduction.

2. Compounds 8-12 have been characterized only through NMR and ESI-MS, which is insufficient for compound purity. Therefore the needs to report the HPLC traces to confirm the formulation of these molecules so that observed cytotoxicity values can be validated.

3. The authors should provide each compound's molecular formula and weight in the experimental section.

4. ESI-MS spectrum for compound 11 is missing from the supplemental file.

5. The authors should provide a table of content in the Supplementary material file.

6. For clarity, the authors should assign the NMR spectrum peaks as inset for all the reported compounds.

Author Response

Point to Point Response to Reviewers´ Comments on Manuscript Molecules - 1876580

A detailed response to the comments and suggestions of reviewer’s 2, that we sincerely thank, is given below. Therein, at first the original referee´s comment is shown in black colour followed by author’s response in blue.

All revisions made to the manuscript have been yellow highlighted.

REVIEWER 2 comments: This paper reported the synthesis and biological evaluation of naphthoquinones which were conjugated with N-acetyl-L-cysteine. Especially, compounds 8 and 9 showed good antitumor activities against the proliferation of tumor cells over normal cells. In the further study, compound 8 could induce the increased ROS formation in different tumor cells with 1 h treatment and induce cell cycle arrest, presumably through the reaction with nucleophiles in cell contents. The topic is interesting and the manuscript is well-organized, while the key issues should be addressed before its publication on Molecules.

Major points:

  1. The antiproliferation activities of compounds 1-4 were missing, as they were performed in the description. The authors are requested to provide these activities as a comparison with the new compounds.

Answer: we thank the reviewer’s remark: in the first version (line 183) we have uncorrectly written 1-4 meaning 8-12.

  1. The positive control in MTT assay was requested to verify the experiment conditions.

Answer: The controls for each cell line are reported in figure 4 (added in revised version)

  1. The secondary cell viability assay (such as cell counting, etc) with different mechanism for compound 8 is required to validate the antiproliferation results, as it was mentioned in the introduction part that these adducts always show deep color, which may affect the detection signal (at 570 nM) of MTT assay.

Answer: In the highly diluted solutions used for biological assays the adducts show no appreciable absorption at 570 nm.

  1. The induction of ROS formation in normal cells by compound 8 is required to be assayed as a control experiment.

Answer: The control has not been performed because IC50 of compound 8 in normal cells is much higher and no detrimental effect has been produced on normal cells viability, as reported in Figure 4.

  1. With different times of compound exposure, the results about the induction of intracellular ROS formation are different. The authors are required to discuss it further.

Answer: The different ROS induction is strictly dependent on the molecular characteristics, this point has been widely discussed in the revised version.

Minor points:

  1. The ‘quinones and amino acids are usually compartmentally separated in living systems’ should be explained further, or the authors should provide the supporting references.

Answer: We provided the supporting reference (21).

  1. The paragraphs about the description of naphthoquinones (below the Figure 2) in the introduction section are suggested to be combined.
  2. The selection of human dermal fibroblast (HDFa) as the normal cell line for toxicity study should be explained, and/or the characteristics of this cell line is suggested to be further described.

Answer: As reported in the literature HDFa are considered a good normal control in cytotoxicity experiments. Indeed, is not possible to find normal controls for all tumor cell lines.

  1. The meaning of R square is suggested to be simply described to increase the readability.

Answer: We the sake of readability, we reported in revised table 2 only IC50 ± SD, as requested in point 7.

  1. In Figure 4, the control column is suggested to be incorporated, or it makes confusions about the columns with significant differences.

Answer: Done

  1. In Figure 5, the resolution of x-/y-axis labels in panel A are too low and the authors are required to adjust them.

Answer: Done

  1. Technical repeats of the MTT experiment should be noted and the IC50s are suggested to be presented in the form of values ± SD.

Answer: Done

  1. The paragraphs about the reaction and characterization in the conclusions section are suggested to be combined.

Answer: We thank the referee and we added a sentence in the conclusion section.

Reviewer 3 Report

This paper reported the synthesis and biological evaluation of naphthoquinones which were conjugated with N-acetyl-L-cysteine. Especially, compounds 8 and 9 showed good antitumor activities against the proliferation of tumor cells over normal cells. In the further study, compound 8 could induce the increased ROS formation in different tumor cells with 1 h treatment and induce cell cycle arrest, presumably through the reaction with nucleophiles in cell contents. The topic is interesting and the manuscript is well-organized, while the key issues should be addressed before its publication on Molecules.

Major points:

1. The antiproliferation activities of compounds 1-4 were missing, as they were performed in the description. The authors are requested to provide these activities as a comparison with the new compounds.

2. The positive control in MTT assay was requested to verify the experiment conditions.

3. The secondary cell viability assay (such as cell counting, etc) with different mechanism for compound 8 is required to validate the antiproliferation results, as it was mentioned in the introduction part that these adducts always show deep color, which may affect the detection signal (at 570 nM) of MTT assay.

4. The induction of ROS formation in normal cells by compound 8 is required to be assayed as a control experiment.

5. With different times of compound exposure, the results about the induction of intracellular ROS formation are different. The authors are required to discuss it further.

Minor points:

1. The ‘quinones and amino acids are usually compartmentally separated in living systems’ should be explained further, or the authors should provide the supporting references.

2. The paragraphs about the description of naphthoquinones (below the Figure 2) in the introduction section are suggested to be combined.

3. The selection of human dermal fibroblast (HDFa) as the normal cell line for toxicity study should be explained, and/or the characteristics of this cell line is suggested to be further described.

4. The meaning of R square is suggested to be simply described to increase the readability.

5. In Figure 4, the control column is suggested to be incorporated, or it makes confusions about the columns with significant differences.

6. In Figure 5, the resolution of x-/y-axis labels in panel A are too low and the authors are required to adjust them.

7. Technical repeats of the MTT experiment should be noted and the IC50s are suggested to be presented in the form of values ± SD.

8. The paragraphs about the reaction and characterization in the conclusions section are suggested to be combined.

Author Response

Point to Point Response to Reviewers´ Comments on Manuscript Molecules - 1876580

A detailed response to the comments and suggestions of reviewer’s 3, that we sincerely thank, is given below. Therein, at first the original referee´s comment is shown in black colour followed by author’s response in blue.

All revisions made to the manuscript have been yellow highlighted.

REVIEWER 3 comments:

Micheletti, Calonghi et al. reported the naphthoquinones-cysteine conjugated compounds synthesis in different experimental conditions and tested their anticancer properties against various cancer cell lines. Compounds, particularly 8-12, have been characterized with the help of spectroscopic techniques such as NMR (1H and 13C) and ESI-MS. The correlation between the increase in reactive oxygen species (ROS) with the help of H2DCFDA and cell cycle analysis to measure the anti-proliferative effect has been evaluated. The author also assisted the toxic impact of respective naphthoquinones-cysteine 8-12 using a non-cancerous human dermal fibroblast (HDFa) cell line. Finally, the observed biological effects were investigated using glutathione, known for forming GSH adducts with RNA/DNA. The mass analysis of the reaction mixture of 8 showed adduct formation, whereas compound 9 remained almost unaffected.

The current manuscript version needs revision before acceptance for publication in the Molecule journal.

  1. Many abbreviations are mentioned in the abstract, such as ROS, MTT, DFCDA, IC50, and SH-SY5Y, which need to be defined in the first place, including in the introduction.

Answer: done

  1. Compounds 8-12 have been characterized only through NMR and ESI-MS, which is insufficient for compound purity. Therefore the needs to report the HPLC traces to confirm the formulation of these molecules so that observed cytotoxicity values can be validated.

Answer: The reviewer suggestion is of course interesting but at this stage there is no time to perform the HPLC analyses. However, the purity of compounds has been evaluated especially by 1H and 13C NMR spectra and for compounds 8, 9, 11, and 12, by melting point range.

  1. The authors should provide each compound's molecular formula and weight in the experimental section.

Answer: we thank the reviewer and added molecular formula and weight in the experimental section.

  1. ESI-MS spectrum for compound 11 is missing from the supplemental file.

Answer: we added ESI-MS spectrum for compound 11 in the supplemental file.

  1. The authors should provide a table of content in the Supplementary material file.

Answer: done

  1. For clarity, the authors should assign the NMR spectrum peaks as inset for all the reported compounds.

Answer: done

Round 2

Reviewer 1 Report

The reviewer is satisfied with all the changes the authors have made to address his comments. Only one minor correction. For the change of "in vitro" in the content, the reviewer meant to italicize it. After correction of this minor issue, this manuscript is recommended for publication. 

Reviewer 2 Report

The revised version of the manuscript addresses most of my comments so that the manuscript can be accepted for publication in Molecules.

Reviewer 3 Report

After revision, the quality of this paper was significantly improved, and could reach the required quality standard for Molecules in my opinion. I suggest accepting it without further revision.